# Robust Prompt Learning for Vision-Language Models with Noisy Labels

## Abstract

Recent advancements in vision-language models (VLMs), designed for simultaneous comprehension of vision and language, have demonstrated significant success in achieving zero-shot classification capabilities. However, despite their impressive performance, it is widely acknowledged that fine-tuning is essential to adapt these models to new target tasks. This adaptation process requires the collection of target datasets, which may introduce incorrect labels and greatly compromise the model performance after fine-tuning. In this paper, our objective is to enhance classification fine-tuning performance by leveraging the zero-shot classification capability under a noisy labeled training dataset. We first conduct a detailed exploration of the behavior of the pre-trained VLMs under various classification text prompts, including human-crafted and LLM-crafted visual characteristics. This investigation reveals that VLMs have tilted knowledge towards some classes, and each prompt exhibits varying expertise for each class. Based on these observations, we introduce a robust training method called PoND, which employs a complementary approach across different types of prompts, leveraging the expertise of each class. We systematically compare the efficacy of the proposed algorithm with existing denoising techniques designed for VLMs and substantiate that our proposed algorithm outperforms prior approaches across 11 real-world datasets.

## 1 Introduction

Despite the proliferation of deep neural networks (DNNs) in various domains, such as image classification He et al. (2016); Dosovitskiy et al. (2020), image generation Goodfellow et al. (2020), and language processing Brown et al. (2020); Touvron et al. (2023a;b), there is a compelling need to explore scenarios that involve multiple modalities. The ability to comprehend various types of inputs simultaneously has driven researchers to develop foundational models, exemplified by vision-language models (VLMs) Radford et al. (2021); Li et al. (2022b). These pre-trained VLMs are well-known for their promising zero-shot performance on various tasks, such as classification and retrieval. However, it is noted that resource-intensive fine-tuning is required to obtain adapted performance in new target domains.

Given the costly nature of tuning all parameters for adaptation of well-constructed pre-trained VLMs, recent research efforts have primarily focused on mitigating adaptation costs Zhou et al. (2022a;b); Khattak et al. (2023a;b). Among these approaches, prompt learning, which involves training a small number of trainable prompt variables with a small number of input samples per class (*e.g.,* up to 16), has garnered significant attention. For example, CoOp Zhou et al. (2022b) improves performance on the target task itself, while others Zhou et al. (2022a); Khattak et al. (2023a;b) focus on improving the generalizability of models to unseen classes.

To effectively implement the aforementioned parameter-efficient fine-tuning of pre-trained VLMs for classification, it is necessary to obtain a training dataset. However, acquiring such a dataset can be expensive and susceptible to noisy labels, as mentioned in various studies Song et al. (2022); Zhang & Sabuncu (2018); Liu et al. (2020); Li et al. (2020b). Despite one of the straightforward methods to address these noisy labels being the use of strong pre-trained zero-shot classification capability Huang et al. (2022) to cleanse the dataset, there have been limited investigations in this area. Only a few studies, such as Wu et al. (2023), have explored the impact of noisy labels on prompt learning, without explicitly utilizing the zero-shot classification capability of VLMs.

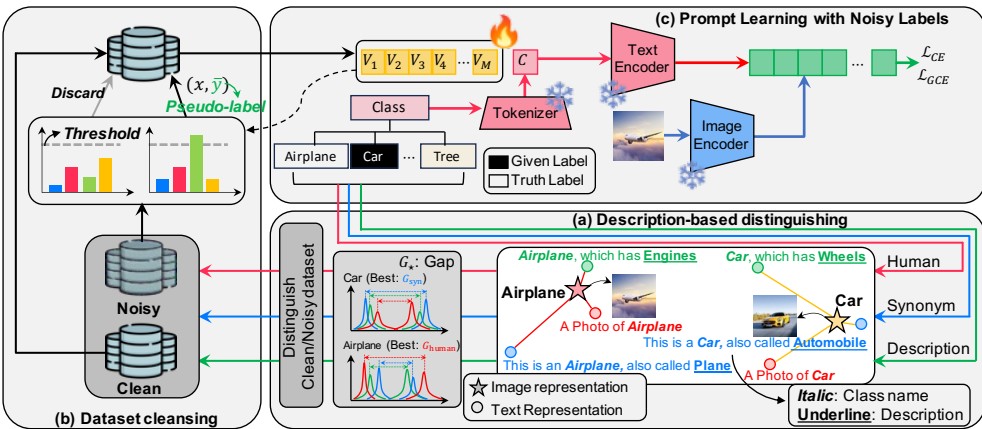

Figure 1: Overview of the proposed method PoND. (a) Distinguishing clean and noisy labels using the best prompt among human-crafted, synonym-based, and description-based prompts. (b) Cleansing the training dataset by relabeling the regarded-as-noisy samples using threshold. (c) Prompt learning via robust loss on the cleansed dataset.

This trend leads us to pose the question: *"What is the proper way of explicitly harnessing the valuable zero-shot classification capability of VLMs for robust training on noisy labels?"* To address this question, we examine the various possible input text prompts, as variations in input prompts demonstrate different zero-shot classification characteristics Menon & Vondrick (2023); Pratt et al. (2023). We evaluate three prompts in total: human-crafted prompt (*e.g.,* 'A photo of **Car**.') Radford et al. (2021), descriptions about target class objects obtained from the external LLMs Brown et al. (2020) as studied in Menon & Vondrick (2023); Pratt et al. (2023) (*e.g.,* 'A **Car** which has wheels'), and using the class word with its synonyms (*e.g.,* 'A photo of **Car**, also called as a Automobile').

In short, each prompt variant has its own advantage regarding each class. As depicted in Figure 1(a), **Car** is well-recognized (described as a shorter distance) when employing a prompt using synonyms, while others (*e.g.,* **Airplane**) are not. Conversely, human-crafted prompt and prompt using descriptions are more (or less) effective for **Airplane** (or **Car**).[1] Building upon these insights, we introduce a novel algorithm, coined PoND, which leverages zero-shot classification capability using various prompts to enhance robustness in the presence of noisy labels.

**Contribution.** We summarize our contributions.

- We investigate zero-shot classification characteristics under various prompts including descriptions obtained from LLMs Menon & Vondrick (2023); Pratt et al. (2023). Additionally, we observe that the synonyms we initially explored also have zero-shot classification capability.

- We find that directly leveraging zero-shot classification capability for cleansing the noisy labels leads to additional incorrect labels, resulting in performance degradation. As an alternative, we search for how to utilize the various prompts for robust training and find they have expertise in per-class aspects to distinguish noisy labels.

- To leverage the zero-shot classification capability of VLMs, we propose a novel robust training method called PoND. The procedure is summarized in Figure 1. In essence, it involves three steps for each iteration. (a) We determine the expert prompt from the set of prompts for each class and categorize the sample into *regarded-as-clean* and *-noisy* sets. (b) We assign pseudo-labels for *regarded-as-noisy* ones whose predicted softmax value is greater than the threshold. (c) The model is trained on the union set of *regarded-as-clean* and pseudo-labeled samples.

- We perform extensive experiments and show the superior performance of PoND compared to the previous method on 11 real-world benchmarks.

---

[1]This is because the text representation obtained from each prompt varies, leading to different expertise levels for each class.

## 2 BACKGROUND

In this section, we briefly summarize preliminaries: classification using VLMs, prompt learning (PL), and zero-shot classification using visual description-based prompts.

**Notations.** Before delving into the preliminary information, we would like to introduce a few notations commonly used in this paper. Firstly, let $\mathcal{D}_{\text{tr}}$ represent the training dataset for a $C$-class classification problem, which comprises pairs of input image $x_i$ and corresponding given label $\hat{y}_i$ denoted as $\{(x_i, \hat{y}_i)\}_{i=1}^{N}$, where the ground truth label of $x_i$ is $y_i$. Here, $y_i$ and $\hat{y}_i \in \{1, \ldots, C\}$, and $N$ represents the total number of training samples. Following prior works, we denote the label $\hat{y}_i$ as *clean* if $\hat{y}_i = y_i$ and *noisy* if $\hat{y}_i \neq y_i$. We refer to the proportion of noisy labels as the *noisy ratio*.

**Classification using the pre-trained VLMs.** VLMs typically consist of two encoders: an image encoder and a text encoder. In the case of CLIP Radford et al. (2021), various CLIP variants incorporate image encoders based on architectures such as ResNet He et al. (2016) or Vision Transformer Dosovitskiy et al. (2020), and text encoders based on the Transformer architecture Vaswani et al. (2017). The primary objective of each encoder is to create embeddings that match a given image and its corresponding text. This matching objective enables the pre-trained CLIP model to be used for various tasks, including classification. The embeddings for the image and text outputs of the CLIP model are formulated as follows:

$$\mathsf{e}_{\text{img}}^{x} = \text{CLIP}_{\text{img}}(x) \quad \mathsf{e}_{\text{txt}}^{c} = \text{CLIP}_{\text{txt}}(\mathcal{T}(\text{CLS}_c)).$$

Here, $\mathcal{T}(\text{CLS}_c)$ represents the text template to make the input prompt; for example, "A photo of $\{\text{CLS}_c\}$," where $\{\text{CLS}_c\}$ denotes the name of the $c^{\text{th}}$ class. This prompt is denoted as $\mathcal{T}_{\text{human}}$ to distinguish it from other prompts. Classification inference is performed by following:

$$\bar{y} = \underset{c \in \{1, \ldots, C\}}{\arg\max} \, P(y = c|x) = \frac{\exp(\cos(\mathsf{e}_{\text{img}}^{x}, \mathsf{e}_{\text{txt}}^{c})/\tau)}{\sum_{i=1}^{C} \exp(\cos(\mathsf{e}_{\text{img}}^{x}, \mathsf{e}_{\text{txt}}^{i})/\tau)}.$$

Here, $\cos(\mathbf{a}, \mathbf{b})$ is the cosine similarity between vectors $\mathbf{a}, \mathbf{b}$. $\tau$ is the temperature hyperparameter.

**Prompt using visual descriptions.** Some recent research has suggested that the text template $\mathcal{T}$ can be expressed by visual descriptions obtained using external knowledge from pre-trained language models, such as GPT-3 Brown et al. (2020), to improve zero-shot classification performance. The details of how these prompts look and how visual descriptions are obtained are explained in Appendix B. In brief, each template for each class receives two inputs, $\mathsf{e}_{\text{txt}}^{c,d} = \text{CLIP}_{\text{txt}}(\mathcal{T}_{\text{vis}}(\text{CLS}_c, \text{DESC}_c^d))$, where $\text{DESC}_c^d$ represents the $d^{\text{th}}$ describing word associated with class $c$, with $d$ ranging from 1 to $D_c$. The value of $D_c$ may vary depending on the class. For example, in the case of Menon & Vondrick (2023), the template $\mathcal{T}_{\text{vis}}$ is "A $\{\text{CLS}_c\}$, which has/have $\{\text{DESC}_c^d\}$." Based on this prompt and visual descriptions, the model infers the class by computing the output as follows:

$$P_{\mathcal{T}_{\text{vis}}}(y = c|x) = \frac{1}{D_c} \sum_{d=1}^{D_c} \frac{\exp(\cos(\mathsf{e}_{\text{img}}, \mathsf{e}_{\text{txt}}^{c,d})/\tau)}{\sum_{k=1}^{C} \exp(\cos(\mathsf{e}_{\text{img}}, \mathsf{e}_{\text{txt}}^{k,d})/\tau)}. \tag{1}$$

**Prompt learning (PL).** As one of the parameter-efficient fine-tuning methods Zhou et al. (2022a;b); Khattak et al. (2023a;b), PL involves setting up a limited number of trainable vectors as prompts while keeping the other inherited encoders frozen. For example, in the case of CoOp Zhou et al. (2022b), it trains $M$ trainable vectors denoted as $\mathcal{V} = [V]_1, \ldots, [V]_M$, where $[V]_m$ has the same dimension as word embeddings (*e.g.,* 512 for CLIP). $\mathcal{V}$ is incorporated into the input of the text encoder:

$$\mathcal{T}_{\text{PL}}(\text{CLS}_c) = [V]_1 \ldots [V]_M [\text{CLS}_c],$$

where $[\text{CLS}_c]$ represents the token value associated with the $c^{\text{th}}$ class word $\{\text{CLS}_c\}$[2]. To train these trainable $\mathcal{V}$, CE loss is typically employed:

$$\mathcal{L}_{\text{CE}}(x, y) = -\sum_{c=1}^{C} \mathbb{1}\{y = c\} \log P(y = c|x). \tag{2}$$

---

[2]Note that, for simplicity, we use the notation $\mathcal{T}_{\text{PL}}(\text{CLS}_c)$ here for token values, even though the previous notation $\mathcal{T}_{\text{vis}}$ is defined for words, not token values.

# 3 UNDERSTANDING THE PRE-TRAINED VLMs

In this section, we first investigate the way of leveraging zero-shot classification capability of VLMs using various prompts to use it for robust training. More precisely, we present two observations: (1) a new type of prompt using synonyms, distinct from the prior visual description approach. We observe that the prompt using synonyms can be used for zero-shot classification, even outperforming human-crafted prompts, and (2) a method of using pre-trained zero-shot classification capability for robust training. In short, directly assigning zero-shot prediction is too dangerous, since zero-shot classification is not sufficiently reliable. Therefore, the zero-shot prediction has to be used carefully.

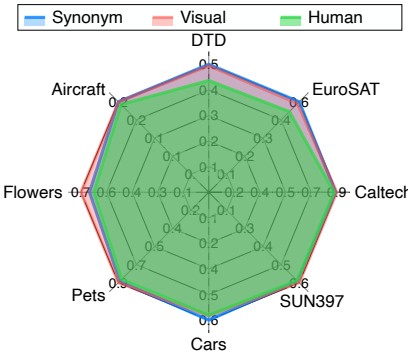

Figure 2: Zero-shot performances under CLIP ViT-B/16.

## 3.1 SYNONYMS FOR ZERO-SHOT CLASSIFICATION

**Why synonyms?** Utilizing synonyms is a form of data augmentation in language processing Zhang et al. (2015). Its fundamental principle is to increase the diversity of the text while preserving semantic information. This aligns with the primary goal of the visual description-based approach. For example, in the case of the class word, "Motorbikes," it can also be described by multiple synonyms, such as {"Bikes", "Scooters", "Two-wheelers", ...}. Therefore, we initially investigate the impact of synonyms on classification using CLIP.

**Obtaining synonyms.** We obtain synonyms of each class-word using LLMs, particularly GPT-3.5-turbo-inst Brown et al. (2020) instead of using word databases, such as WordNet Feinerer & Hornik (2023), for covering the fine-grained tasks, *e.g.,* A310 in Aircraft Maji et al. (2013). We construct the LLM-prompt as follows:

> *Q: What are the synonyms of* {CLS}?
> *A: There are several synonyms of* {CLS}:

When {SYN} denotes the synonyms and {$ANT_c$} is one of the class-words other than {$CLS_c$}, *i.e.,* {$CLS_{c'}$} where $c' \in [C] \setminus \{c\}$, $\mathcal{T}_{syn}$ is:

> *This is a photo of* {CLS}, *which is also called as a* {SYN}. *It is not a* {ANT}.

By using the above prompt, we classify the class using Eq. (1) with replacement of $\mathcal{T}_{vis}$ to $\mathcal{T}_{syn}$.

**$\mathcal{T}_{syn}$ can be used for zero-shot classification.** We evaluate the new prompt to see if it can be used for zero-shot classification. As shown in Figure 2, synonym-based classification exhibits improved accuracy compared to the $\mathcal{T}_{human}$ in several benchmarks. It also shows a performance similar to that of $\mathcal{T}_{vis}$. This test accuracy indicates its ability can be considered as a candidate to help the robust training, along with $\mathcal{T}_{human}$ and $\mathcal{T}_{vis}$.

## 3.2 WAYS TO USE ZERO-SHOT CLASSIFICATION

From the sufficient zero-shot classification capability of VLMs, the remaining question is how to use it for robust training under noisy labels. Simply speaking, we can use the knowledge for robust training in two ways: (1) distinguishing between clean and noisy labels, and (2) labeling given images using the inference results. Hereinafter, we investigate these cases in detail.

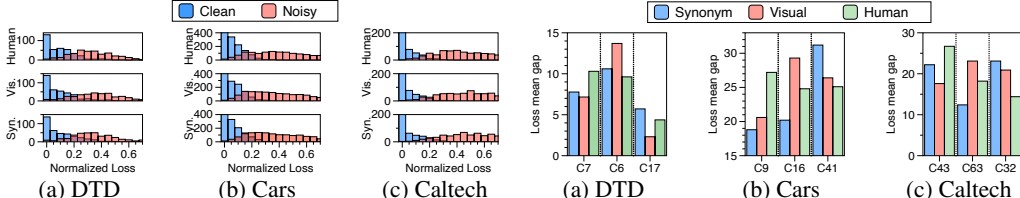

Figure 3: Normalized loss histogram of $50\%$ symmetric noisy case with ViT-B/16 model.

Figure 4: The gap values between the mean loss of clean and noisy samples for each class.

**VLMs have distinguishability of noisy samples.** First, we evaluate VLMs' distinguishability of noisy labels. To investigate this, we measure and present the normalized loss histogram in Figure 3. In all cases, each prompt demonstrates an adequate capability in identifying clean samples, which have a lower loss compared to noisy ones.

**Expertise of each prompt in specific classes.** The remaining question concerns the approach to utilizing various prompts. To evaluate their characteristics, we measure the gap between the mean loss values of clean and noisy sets for each class. As illustrated in Figure 4, each prompt exhibits a specialty in certain classes. For instance, in the case of $41^{\text{st}}$ class on the Stanford Cars dataset, $\mathcal{T}_{\text{syn}}$ shows better distinguishability compared to other prompts. Conversely, $\mathcal{T}_{\text{vis}}$ demonstrates stronger distinguishability in the $9^{\text{th}}$ class. Therefore, to effectively use multiple prompts to find noisy samples, we have to use one of the most suitable prompts for each class.

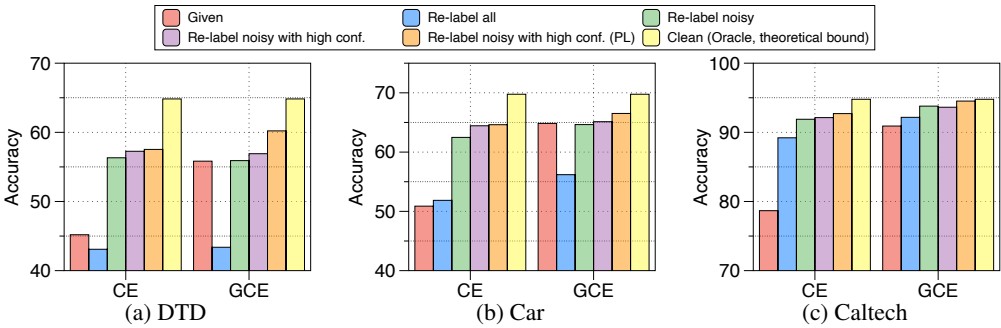

Figure 5: Performance on different labeling methods using the pre-trained knowledge. Here, we utilize oracle distinguishing information (except for Given and Re-label all) to verify the re-labeling impact. Here, Clean is not practically achievable.

**Better way of labeling using VLMs.** To verify the best criteria for obtaining labels via VLMs under noisy conditions, we examine six possible cases and conduct analysis: (1) Using the given label $\hat{y}$ without re-labeling, (2) Replace $\hat{y}$ to $\bar{y}$ which is the prediction using $\mathcal{T}_{\text{human}}$, (3) Change $\hat{y} \neq y$ only to $\bar{y}$ (It is practically impossible to distinguish but we give additional information for exploration), (4) Change $\hat{y} \neq y$ to $\bar{y}$ whose predictions are sufficiently confident ($\max_c P_{\mathcal{T}_{\text{human}}}(y = c|x) > 0.95$), (5) Change $\hat{y} \neq y$ to $\bar{y}$ whose $\max_c P_{\mathcal{T}_{\text{PL}}}(y = c|x) > 0.95$, and (6) the oracle clean case. As indicated in Figure 5, when we directly assign the inference labels to each sample, it can drop the performance (See (2) of DTD). The most promising labeling way is using $\mathcal{T}_{\text{PL}}$ (See (5) for all cases). This is because PL can adapt to the target task, while the other zero-shot-based approach cannot. Therefore, when we re-assign the labels to the regarded-as-noisy samples, $\mathcal{T}_{\text{PL}}$ has to be used.

### 3.3 OBSERVATION SUMMARY

The summary of our findings is as follows: **(Obs 1):** A synonym can serve as a good candidate for zero-shot classification using VLMs. **(Obs 2):** The pre-trained VLMs have a good distinguishability of noisy labels from the given probably noisy labeled dataset. Moreover, each prompt has its own advantage for each class. **(Obs 3):** To assign the cleansed labels from the VLMs, re-labeling samples whose confidence (*e.g.*, max-softmax) is larger than threshold under $\mathcal{T}_{\text{PL}}$ is the most reliable re-labeling method. Based on these observations, we have developed an algorithm, and details are provided in the following section.

**Algorithm 1:** `Set_Distinguishing`

**Input:** $\mathcal{D}_{\text{tr}}$, GMM thresh. $g$, Prompt set $\mathcal{T}_{\text{set}}$
**Initialize:** $\hat{\mathcal{D}}_{\text{cl}} = \emptyset, \hat{\mathcal{D}}_{\text{no}} = \emptyset$
**for** $c = 1, ..., C$ **do**
    # Initialize $G$ for at least one $\mathcal{T}$ is used
    $G = 0$
    # Output is a Gaussian dist. of the best $\mathcal{T}$
    **for** $\mathcal{T} \in \mathcal{T}_{set}$ **do**
        # Measure CE loss for all samples, Eq. (2)
        $L = \{\ell_i | \ell_i = \mathcal{L}_{\text{CE}}(x_i, y_i), y_i = c\}$
        # Run GMM-estimation, Eq. (3)
        $p_1, p_2 = \text{GMM}(L, \mathcal{T})$
        # Select the best prompt, Gaussian dist.
        **if** $|\mu_1 - \mu_2| > G$ **then**
            $i = \arg\min_{i \in \{1,2\}} \mu_i$
            $G = |\mu_1 - \mu_2|$ and $p = p_i$
        **end**
    **end**
    # Select clean/noisy sets
    $D_{\text{cl}} = \{(x, y) | p(\ell) > g, (x, y) \in \mathcal{D}_{\text{tr}}\}$
    $D_{\text{no}} = \{(x, y) | p(\ell) \le g, (x, y) \in \mathcal{D}_{\text{tr}}\}$
    # Update aggregated clean/noisy sets
    $\hat{\mathcal{D}}_{\text{cl}} = \hat{\mathcal{D}}_{\text{cl}} \cup D_{\text{cl}}, \hat{\mathcal{D}}_{\text{no}} = \hat{\mathcal{D}}_{\text{no}} \cup D_{\text{no}}$
**end**
**Output:** $\hat{\mathcal{D}}_{\text{cl}}, \hat{\mathcal{D}}_{\text{no}}$

**Algorithm 2:** `Re-Labeling`

**Input:** $\hat{\mathcal{D}}_{\text{no}}$, Re-labeling threshold $\alpha$
# Pseudo-labeling for confident samples
$\bar{\mathcal{D}} = \{(x_i, \bar{y}_i) | \bar{y}_i = \arg\max_c P_{\text{PL}}(y = c | x_i)$

$\forall (x_i, y_i) \in \hat{\mathcal{D}}_{\text{no}} \text{ and } \max_c P_{\text{PL}}(y = c | x_i) > \alpha\}$
**Output:** Cleansed noisy set $\bar{\mathcal{D}}$

**Algorithm 3:** `PoND`

**Input:** $\mathcal{D}_{\text{tr}}$, Re-labeling threshold $\alpha$, GMM
        threshold $g$, Description set $\mathcal{T}_{\text{set}}$,
        Iteration $T$, GCE parameter $k$.
**Initialize:** $\mathcal{T}_{\text{PL}}, \mathcal{V}$
**for** $t = 1, ..., T$ **do**
    # Clean/noisy set select via descr. (Alg. 1)
    $\hat{\mathcal{D}}_{\text{cl}}, \hat{\mathcal{D}}_{\text{no}} =$
      `Set_Distinguishing`$(\mathcal{D}_{\text{tr}}, \mathcal{T}_{\text{set}}, g)$
    # Pseudo-labeling Noisy-set (Alg. 2)
    $\bar{\mathcal{D}}_{\text{no}} = $ `Re-Labeling`$(\hat{\mathcal{D}}_{\text{no}}, \alpha)$
    # Construct train set
    $\mathcal{D} = \hat{\mathcal{D}}_{\text{cl}} \cup \bar{\mathcal{D}}_{\text{no}}$
    # Train using GCE loss, Eq. (4)
    Update $\mathcal{V}$ (incl. $\mathcal{T}_{\text{PL}}$) on $\mathcal{D}$ using $\mathcal{L}_{\text{GCE}}$
**end**

# 4 PROPOSED METHOD: PoND

In this section, we describe the robust prompt learning method under noisy labels combining prompts.

**Overview.** Our method consists of three steps in each iteration. First, we select clean samples using a two-cluster Gaussian mixture model (GMM) constructed on the losses obtained from the prompts. Here, we select the prompt with the best discriminative performance between the *regarded-as-clean* and *-noisy* sets from among the prompts (based on **Obs 2**), including the synonym-based one (based on **Obs 1**). After selecting the regarded-as-noisy samples, we generate pseudo-labels using the predictions of prompt learning, rather than using the zero-shot classification results, to prevent additional generation of noisy samples (based on **Obs 3**), and select highly confident samples so that they can contribute to the training. The proposed algorithm, coined PoND (**P**rompt learning **o**n **N**oisy labels through **D**enoising using various prompts), is described in Algorithm 3.

**Module 1: Set distinguishing.** For each training iteration, we first distinguish the set into the regarded-as-clean $\hat{\mathcal{D}}_{\text{cl}}$ and -noisy $\hat{\mathcal{D}}_{\text{no}}$. We run GMM estimation, which is summarized as follows:

$$p_1(\ell; \mu_1, \sigma_1), p_2(\ell; \mu_2, \sigma_2) = \text{GMM}(L_c, \mathcal{T}) \tag{3}$$

where $\mathcal{T} \in \mathcal{T}_{\text{set}}$ and $L_c$ is the set of per-sample losses for class $c$, and $p_1$ and $p_2$ are two estimated Gaussian distributions whose mean values are $\mu_1$ and $\mu_2$, respectively, and $\mathcal{T}_{\text{set}} = \{\mathcal{T}_{\text{human}}, \mathcal{T}_{\text{vis}}, \mathcal{T}_{\text{syn}}, \mathcal{T}_{\text{PL}}\}$. We select the best prompt from the set of given prompts $\mathcal{T}_{\text{set}}$ to leverage the expertise of each prompt for each class (**Obs 2**).

Among the four prompt candidates, the best prompt under interest is to find the most distinguishable prompt for each class. Therefore, we compute the gap between $\mu_1$ and $\mu_2$, *i.e.*, $G = |\mu_1 - \mu_2|$. The intuition here is that a larger gap between the two mean values indicates a better distinguishability between noisy samples and clean samples. In other words, the selected prompt is considered an expert in that class. Furthermore, we also utilize $\mathcal{T}_{\text{PL}}$, which evolves as the training progresses, so that it can facilitate a smooth transition from pre-trained knowledge to adapted knowledge. `Set_Distinguishing` is described in Algorithm 1.

| | S 0.25 | S 0.5 | S 0.75 | A 0.3 | S 0.25 | S 0.5 | S 0.75 | A 0.3 | S 0.25 | S 0.5 | S 0.75 | A 0.3 | S 0.25 | S 0.5 | S 0.75 | A 0.3 |
|---|---|---|---|---|---|---|---|---|---|---|---|---|---|---|---|---|
| | | | | | | | | | | | | | | | | |
| **Caltech-101** (RN50 / ViT-L/32) | | | | | | | | | **Flowers** (RN50 / ViT-L/32) | | | | | | | |
| Vanilla | 86.76 | 71.49 | 53.03 | 58.87 | 87.37 | 78.67 | 60.43 | 60.29 | 81.84 | 70.84 | 44.31 | 64.99 | 84.28 | 73.97 | 48.70 | 64.01 |
| PTNL | 91.19 | 87.48 | 69.64 | 88.99 | 94.81 | 90.91 | 75.55 | 92.65 | 87.24 | 84.41 | 73.73 | 77.99 | 87.42 | 83.65 | 77.00 | 81.17 |
| Ours | **92.76** | **91.21** | **88.41** | **90.98** | **95.14** | **94.43** | **93.36** | **94.17** | **87.47** | **85.44** | **78.40** | **81.12** | **88.47** | **87.62** | **81.47** | **83.09** |
| **DTD** | | | | | | | | | **Pets** | | | | | | | |
| Vanilla | 55.12 | 44.80 | 23.87 | 41.06 | 56.71 | 45.19 | 25.15 | 42.33 | 80.20 | 71.23 | 43.32 | 60.62 | 83.12 | 73.85 | 44.84 | 62.27 |
| PTNL | 60.33 | 56.28 | 39.24 | 52.48 | 62.18 | 55.83 | 41.84 | 54.44 | 87.66 | 84.79 | 71.07 | 81.95 | 89.22 | 85.59 | 73.43 | 84.45 |
| Ours | **60.93** | **57.77** | **47.02** | **54.76** | **63.12** | **59.49** | **48.38** | **56.23** | **88.58** | **85.38** | **77.47** | **85.06** | **90.21** | **87.77** | **81.71** | **88.21** |
| **EuroSAT** | | | | | | | | | **Aircraft** | | | | | | | |
| Vanilla | 71.12 | 56.48 | 29.19 | 50.22 | 73.00 | 59.22 | 32.19 | 50.37 | 23.33 | 18.60 | 10.52 | 17.56 | 24.63 | 20.02 | 12.61 | 18.89 |
| PTNL | 74.29 | 62.47 | 30.26 | 44.52 | 73.91 | 66.63 | 39.40 | 50.55 | 25.53 | 23.29 | 14.69 | 21.87 | 27.86 | 24.00 | 18.93 | 23.63 |
| Ours | **75.17** | **67.86** | **40.84** | **53.05** | **74.58** | **69.28** | **46.07** | **54.81** | **27.03** | **23.68** | **19.30** | **22.22** | **28.73** | **26.52** | **21.86** | **25.22** |
| **Cars** | | | | | | | | | **SUN397** | | | | | | | |
| Vanilla | 56.10 | 46.28 | 28.02 | 41.44 | 56.78 | 50.88 | 33.61 | 43.75 | 66.00 | 67.06 | 54.84 | 65.99 | 72.81 | 67.49 | 54.89 | 69.74 |
| PTNL | 63.27 | 61.26 | 56.02 | 58.69 | 67.30 | 64.84 | 60.08 | 62.54 | 68.52 | 67.69 | 54.89 | 68.99 | 72.51 | 69.14 | 66.34 | 72.40 |
| Ours | **64.13** | **62.50** | **57.28** | **59.30** | **67.51** | **66.36** | **61.19** | **63.54** | **69.09** | **67.74** | **67.18** | **70.32** | **73.35** | **70.70** | **67.64** | **74.12** |
| **Food101** | | | | | | | | | **ImageNet** | | | | | | | |
| Vanilla | 72.81 | 62.50 | 45.04 | 56.41 | 76.00 | 64.07 | 45.16 | 54.20 | 60.18 | 57.57 | 49.34 | 49.79 | 63.69 | 62.67 | 56.75 | 65.04 |
| PTNL | 77.61 | 75.38 | 69.90 | 74.76 | 80.48 | 76.33 | 71.55 | 78.27 | 61.22 | 60.53 | 57.57 | 51.17 | 66.12 | 65.50 | 63.84 | 65.66 |
| Ours | **78.59** | **77.34** | **75.23** | **76.22** | **81.44** | **78.33** | **76.78** | **81.05** | **61.90** | **61.60** | **59.13** | **60.10** | **67.58** | **66.69** | **64.51** | **66.66** |
| **UCF101** | | | | | | | | | **Average** | | | | | | | |
| Vanilla | 67.95 | 59.89 | 43.45 | 49.00 | 67.95 | 60.90 | 44.45 | 52.56 | 65.58 | 56.98 | 38.63 | 50.54 | 67.85 | 59.72 | 41.71 | 53.04 |
| PTNL | 71.59 | 68.17 | 63.32 | 66.18 | 76.08 | 69.61 | 65.36 | 71.27 | 69.86 | 66.52 | 55.43 | 62.51 | 72.54 | 68.37 | 59.39 | 67.00 |
| Ours | **72.31** | **70.39** | **64.99** | **68.53** | **77.51** | **71.81** | **65.51** | **75.49** | **70.72** | **68.26** | **61.39** | **65.61** | **73.42** | **70.82** | **64.41** | **69.33** |

Table 1: Results on 11 benchmarks using ResNet-50 (RN50) and ViT-L/32 CLIP. The results are the average of 10 random seeds, and the best are highlighted in **bold**.

**Module 2: Re-labeling noisy labels.** For the next step, we need to include the samples in $\hat{\mathcal{D}}_{no}$ in the training procedure to increase the information during training. The most important criterion here is to avoid generating additional noisy labels. Therefore, we employ a confidence-based re-labeling method using the trained prompt $\mathcal{T}_{PL}$. This is because we want to avoid increasing the noisy ratio generated by biased pre-trained knowledge (**Obs 3**). As described in Algorithm 2, after obtaining $\hat{\mathcal{D}}_{no}$, samples whose max-softmax outputs exceed the threshold hyperparameter $\alpha$ are assigned the model prediction to form relabeled $\bar{\mathcal{D}}_{no}$. Ultimately, we create the training dataset $\mathcal{D} = \hat{\mathcal{D}}_{cl} \cup \bar{\mathcal{D}}_{no}$.

**Entire training procedure using GCE loss.** After having $\mathcal{D}$ at the beginning of each training epoch, we optimize the prompt $\mathcal{V}$. To reduce the impact of possibly remaining noisy labels, we use the GCE loss Zhang & Sabuncu (2018) defined as:

$$\mathcal{L}_{GCE}(x,y) = \frac{1}{C} \sum_{c=1}^{C} \frac{1 - (P_{\mathcal{T}_{PL}}(y=c|x))^k}{g}, \quad (4)$$

where $k$ is the GCE hyperparameter. Our entire training procedure is described in Algorithm 3.

## 5 EXPERIMENT

In this section, we describe the experimental results on several benchmarks and provide analysis.

### 5.1 EXPERIMENTAL SETTINGS

**Datasets.** We conduct experiments on diverse datasets, which are use in CoOp. We used 11 datasets: EuroSAT Helber et al. (2019), Cars Krause et al. (2013), SUN397 Xiao et al. (2010), Pets Parkhi et al. (2012), Food101 Bossard et al. (2014), DTD Cimpoi et al. (2014), UCF101 Soomro et al. (2012), Flower102 Nilsback & Zisserman (2008), Aircraft Maji et al. (2013), Caltech101 Fei-Fei et al. (2004), and ImageNet Russakovsky et al. (2015). Detailed explanations for each dataset are provided in Appendix C.

**Models and comparison baselines.** Among various CLIP types, we compare two different architectures: ResNet-50 (RN50) and ViT-L/32. We compare our method with CoOp, denoted as Vanilla, and PTNL Wu et al. (2023). Implementation details are described in Appendix C.

**Implementation details.** We follow the implementation of CoOp Zhou et al. (2021) and PTNL Wu et al. (2023). Specifically, we use the front-prompt, which means $\mathcal{V}$ is set in front of the class words, and the prompt is shared among classes. For each class, 16 samples are given, and the noisy ratio represents the portion of noisy labels. For a deeper understanding, we check both symmetric (denoted as S) and asymmetric (denoted as A) noisy types (See Appendix D). We train for 50 epochs with a batch size of 32 and leverage the SGD optimizer with a momentum of 0.9. The initial learning rate

is 0.002 and cosine-annealing scheduler is used. We set the hyperparameters for GCE $k = 0.5$ and GMM $g = 0.5$, following prior works, and simply select the re-labeling parameter $\alpha = 0.95$.

## 5.2 RESULTS

**Overall results.** To begin with, as indicated in Table 1, the proposed method demonstrates superior performance across all datasets and noisy configurations compared to both the previous robust training method and the vanilla model. For example, in the RN50 case, PoND shows an increase of 22.76% and a 5.96% increment compared to the vanilla and PTNL performance, respectively, on the average of 11 datasets when 75% of labels are symmetrically flipped. Both symmetric and asymmetric cases, PoND shows superiority to the others.

More precisely, the performance improvement from PTNL is significant when the noise ratio is high, *i.e.,* when the noise ratio increases from 25% to 75%, the improvement gap increases from 0.88% to 5.02% in the ViT case. This is due to the fundamental nature of GCE. As argued in Zhang & Sabuncu (2018), GCE ignores the regarded-as-noisy samples by adapting to MAE loss (whose loss value is slighter than CE), while regarded-as-clean samples incur loss from CE loss. On the other hand, PoND leverages noisy label samples after cleansing them, while PTNL does not. This phenomenon is also observable in the asymmetric case, which has severe noise in some classes.

## 5.3 ANALYSIS

For a deeper understanding of PoND, we provide additional analysis results. Here, for checking the sensitivity of PL options, we first explore various options, such as the size of trainable prompts and the number of samples in each class. We then conduct an ablation study to verify the impact of each component. Finally, we describe the impact of each prompt in $\mathcal{T}_{\text{set}}$ defined in Section 4. All experiments are conducted using Caltech101, EuroSAT, and Oxford Flowers datasets under a symmetric noise case with a

| Setting | Configuration | Caltech-101 | EuroSAT | Flowers |
|---|---|---|---|---|
| | Front | 91.16/94.32 | 62.09/63.74 | 84.91/86.40 |
| Class-word position | Middle | 90.51/93.65 | 66.15/68.42 | 85.97/86.80 |
| | End | 91.21/94.43 | 67.86/69.28 | 85.44/87.62 |
| | 1 | 90.40/91.56 | 62.62/68.30 | 72.63/78.56 |
| | 2 | 90.64/92.19 | 62.04/68.80 | 78.52/78.68 |
| Size of $\mathcal{V}$ | 4 | 90.91/92.59 | 63.78/68.78 | 85.03/84.57 |
| | 8 | 91.04/94.32 | 63.43/68.85 | 85.22/86.19 |
| | 16 | 91.21/94.43 | 67.86/69.28 | 85.44/87.62 |
| | 2 | 88.11/92.90 | 35.47/53.02 | 69.91/69.31 |
| Number of shots | 4 | 89.18/93.47 | 36.15/53.21 | 72.55/77.43 |
| | 8 | 90.30/94.32 | 53.54/56.02 | 76.61/81.12 |
| | 16 | 91.21/94.43 | 67.86/69.28 | 85.44/87.62 |
| Sharing of $\mathcal{V}$ among classes | Share | 91.21/94.43 | 67.86/69.28 | 85.44/87.62 |
| | Not-share | 87.42/92.66 | 70.59/72.26 | 89.52/92.85 |

Table 2: The performances on various PL settings. We only change the setting and configuration from the case mentioned in the implementation part.

0.5 noise ratio. We report RN50 and ViT performances in the RN50/ViT order. Please refer to the further analysis in Appendix.

**Various PL configurations.** PL exhibits several implementation options, including the size of trainable prompt $\mathcal{V}$, the position of the class word, the number of given images per class, and whether the prompt is shared or not. We verify the consistency of PoND in various cases, as described in Table 2. Firstly, when the class word is placed at the end of the prompt, it generally shows slightly better performance than the others. Secondly, when the prompt size is reduced to 1 from 16, the performance drops but not significantly. This phenomenon is also observed in other research Bang et al. (2023). However, the number of shots has a significant impact on performance. When the number of shots is 2, which means only one sample for each class is correct and the other one is incorrect, the performance drops significantly compared to the 16 case. This suggests that obtaining a greater number of samples is crucial. Finally, there is no tendency for the existence of a per-class prompt for each dataset, which is also aligned with CoOp Zhou et al. (2022b).

**Ablation study.** We assess the influence of each component by conducting an ablation study. The primary components of PoND are: (1) Dividing $\mathcal{D}_{\text{tr}}$ into $\hat{\mathcal{D}}_{\text{clean}}$ and $\hat{\mathcal{D}}_{\text{noisy}}$. Without this module, we would have to use the entire dataset, with or without GCE loss. (2) Pseudo-labeling through thresholding, which involves selecting confidently predicted samples to enhance robustness. Without this step, all inference would have to

| Configuration | | | Caltech-101 | EuroSAT | Flowers |
|---|---|---|---|---|---|
| Alg 1. | Alg.2 | $\mathcal{L}_{\text{GCE}}$ | | | |
| ✗ | ✗ | ✗ | 71.49/78.67 | 56.48/59.22 | 70.84/73.97 |
| $\mathcal{O}$ | ✗ | ✗ | 74.20/85.88 | 58.49/62.87 | 78.60/82.10 |
| $\mathcal{O}$ | $\mathcal{O}$ | ✗ | 80.69/88.48 | 60.73/64.32 | 77.99/82.26 |
| ✗ | ✗ | $\mathcal{O}$ | 87.48/90.91 | 62.47/66.63 | 84.41/83.65 |
| $\mathcal{O}$ | ✗ | $\mathcal{O}$ | 88.92/93.83 | 64.42/67.54 | 84.87/86.35 |
| $\mathcal{O}$ | $\mathcal{O}$ | $\mathcal{O}$ | 91.21/94.43 | 67.86/69.28 | 85.44/87.62 |

Table 3: Ablation study of Alg 1, Alg 2 and GCE.

be utilized. As outlined in Table 3, selecting clean samples can enhance robustness, and confident labeling also contributes to training. Moreover, employing GCE loss enables PoND to bolster robustness further.

**Description-configuration analysis.** We describe various combinations of prompts for $\mathcal{T}_{\text{set}}$ in Table 4. When we employ multiple prompts for $\mathcal{T}_{\text{set}}$, the performance improves. For example, the best performance achieved by using one prompt for the Caltech dataset in the ViT case is 93.98%, while the lowest case with two prompts shows 94.09%. It suggests that PoND effectively utilizes the expertise of each prompt in distinguishing clean samples.

| Used prompt | | | Caltech-101 | EuroSAT | Flowers |
|---|---|---|---|---|---|
| Human | Vis. | Syn. | | | |
| $\mathcal{O}$ | ✗ | ✗ | 89.91/93.80 | 65.63/67.99 | 85.16/84.98 |
| ✗ | $\mathcal{O}$ | ✗ | 89.98/93.85 | 63.00/66.94 | 84.75/86.58 |
| ✗ | ✗ | $\mathcal{O}$ | 90.08/93.89 | 65.83/68.86 | 84.78/85.44 |
| ✗ | $\mathcal{O}$ | $\mathcal{O}$ | 90.71/94.18 | 66.48/68.94 | 84.80/86.92 |
| $\mathcal{O}$ | ✗ | $\mathcal{O}$ | 90.20/94.09 | 66.94/69.01 | 85.10/86.55 |
| $\mathcal{O}$ | $\mathcal{O}$ | ✗ | 90.52/93.98 | 67.12/69.11 | 84.98/86.64 |
| $\mathcal{O}$ | $\mathcal{O}$ | $\mathcal{O}$ | 91.21/94.43 | 67.86/68.28 | 85.44/87.62 |

Table 4: Performance analysis when different combination of prompt set $\mathcal{T}_{\text{set}}$ is given.

## 6    HYPERPARAMETER SENSITIVITY

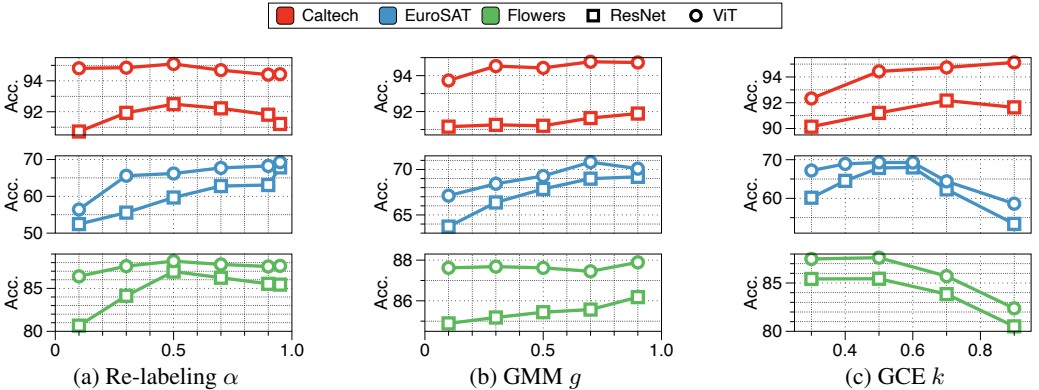

(a) Re-labeling $\alpha$                (b) GMM $g$                (c) GCE $k$

Figure 6: Hyperparameter sensitivity analysis

**Hyperparameter sensitivity.** We primarily utilize three hyperparameters: GCE $k$, GMM $g$, and the re-labeling threshold $\alpha$. The results of parameter sensitivity are written in Figure 6. Regarding the parameter $k$, the Caltech dataset shows improved performance for values larger than our primary experiment setting of 0.5, identical to the PTNL setting Wu et al. (2023). Conversely, the Flower dataset exhibits superior performance at lower settings. From a re-labeling parameter $\alpha$ perspective, sensitivity is not markedly significant; however, EuroSAT demonstrates enhanced performance as the threshold increases. This improvement is ascribed to the fact that the lower initial performance of EuroSAT tends to magnify the ratio of noisy labels when utilizing inference output. Finally, regarding the GMM parameter $g$, it is observed that a higher GMM threshold generally yields better performance, even though 0.5 is employed in this study. Overall, even though we use relatively simplistic hyperparameters, which are not fully tuned but inherited from prior works, further tuning of the parameters could lead to even greater performance enhancements.

## 7    RELATED WORK

**Vision-language models.** Before the emergence of CLIP, models like Lu et al. (2019); Das et al. (2017); De Vries et al. (2017); Qi et al. (2020); Gan et al. (2020); Yu et al. (2021); Li et al. (2020a) had made contributions in this area. However, the introduction of CLIP Radford et al. (2021) in 2021 marked a significant breakthrough. Building on this, ALIGN Jia et al. (2021) followed a similar training approach and ALBEF Li et al. (2021) introduced multi-modal transformer operations to handle both image and text information in an aggregated manner. BLIP Li et al. (2022b; 2023) took a generative approach, capable of generating captions for input images. LiT Zhai et al. (2022) focused on enhancing training efficiency through selective parameter updates and FILIP Yao et al. (2021) addressed fine-grained training. Florence Yuan et al. (2021) expanded representation learning to cover video. Recent research efforts emphasize grounding information, as proposed in Rasheed et al. (2023).

Additionally, there is a growing interest in guiding input images, exemplified by the SoM Yang et al. (2023) using GPT-4V OpenAI (2023).

**Description for VLMs classification.** To harness pre-trained knowledge for zero-shot classification, Menon & Vondrick (2023) presented that they used a GPT model to obtain the visual characteristics of specific target classes, and Pratt et al. (2023) extended this idea by employing multiple prompts to extract characteristics for each class.

**Prompt learning.** PL initially emerged in the realm of NLP tasks and later found application in VLMs. CoOp Zhou et al. (2022b) was among the pioneers to directly employ prompt learning in VLMs. Subsequently, the same research group extended their work to CoCoOp Zhou et al. (2022a) for handling novel classes. MaPLe Khattak et al. (2023a) introduced a variant of prompt learning that optimizes both image and text perspectives simultaneously. In PromptSRC Khattak et al. (2023b), it was observed that prompt learning could lead to the forgetting of valuable, pre-trained generalizable knowledge. Thus, self-regularization techniques were developed to prevent this. Additionally, various studies have explored enhancing PL under active learning Bang et al. (2023) and addressing backdoor attacks Bai et al. (2023).

**Robust loss for learning with noisy labels.** For robust training on noisy labels, Wang et al. (2019) introduced symmetric CE loss, combining it with reverse CE loss. GCE Zhang & Sabuncu (2018) reduced the influence of noisy labels. ELR Liu et al. (2020) tackled the issue of noisy label memorization, and ALASCA Ko et al. (2022) introduced label smoothing. Recently, Cheng et al. (2023) proposed a representation-based regularizer to prevent memorization.

**Semi-supervised approach for LNL.** DivideMix Li et al. (2020b) is proposed to use two networks to generate complementary pseudo-labels using the MixMatch algorithm. Karim et al. (2022) focused on improving class balance in semi-supervised-based training, while Kim et al. (2021) proposed FINE to detect noise labels on embedding dimension. Li et al. (2022a) and Li et al. (2022c) proposed noisy label selection and cleansing algorithms based on neighborhood and similarity scores, respectively. Additionally, in Xia et al. (2022), an uncertainty-based method was introduced.

**Other robust training methods for LNL.** From another standpoint, the C2D Zheltonozhskii et al. (2022) approach asserted that initiating training from pre-trained models, especially contrastive learning models, yields superior results compared to previous methods. In Ko et al. (2023) and Ahn et al. (2023), authors also leveraged pre-trained large models to identify noisy labels by freezing feature extractors. Ortego et al. (2021) proposed a robust training method from a multi-view perspective. Additionally, Optimal Transport-based approaches Xia et al. (2022); Feng et al. (2023); Chang et al. (2023) have emerged in the past two years. Similar to our approach, PTNL Wu et al. (2023) argued that the GCE loss is an effective choice when applying prompt learning to VLMs in the presence of noisy labels. Before the above works, addressing noisy labels in training datasets has been a significant research area, especially in the realm of deep learning Song et al. (2022). Prior to 2022, numerous studies Xiao et al. (2015); Lee et al. (2018); Northcutt et al. (2021); Bahri et al. (2020); Wang et al. (2019); Han et al. (2018); Yu et al. (2019); Cheng et al. (2021); Ma et al. (2020); Zhou et al. (2021); Zheng et al. (2020); Jindal et al. (2016); Lee et al. (2019) have sought ways to mitigate the impact of noisy labels during training.

# 8 CONCLUSION

In this paper, we present an innovative approach to train vision language models (VLMs) with robustness, especially when dealing with noisy labels in the training dataset. Our approach comprises two key components: (1) Splitting the provided samples into two categories, namely those considered clean and those identified as noisy, using the description that exhibits the highest expertise in each class. (2) Assigning pseudo-labels to the samples with sufficiently high confidence. These procedures are built upon our findings that different approaches to classifying samples under VLMs can excel in various class expertise, and on-training-based pseudo-labeling is the most dependable method. Through extensive experimentation across diverse datasets and architectures, we demonstrate the effectiveness of our proposed method compared to existing approaches, including VLM-based robust training and training from scratch methods.

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

---

-Supplementary Material-

# Robust Prompt Learning for Vision-Language Models with Noisy Labels

---

This supplementary material provides additional analysis and explanation of our paper, "Robust Prompt Learning for Vision-Language Models with Noisy Labels", which were not included in the main manuscript due to page constraints. First of all, for the readers' better better understanding, we describe the notations used in the main manuscript in Appendix A. Appendix B details how descriptions are obtained, including a summary of prior works. Appendix C outlines the characteristics of each dataset and implementation details. Appendix D describes the noisy generation method. For further analysis, we illustrate the selected prompt during training in Appendix E. We present cases of other prompt learning methods in Appendix F, which aim to enhance prompt learning with generalizability. In Appendix G, H, I, we deliver further analysis about the proposed algorithm.

## A    NOTATION

| Notation | Description |
|---|---|
| $\mathcal{T}_{\text{set}}$ | Template set |
| $\mathcal{T}_{\text{human}}$ | Human-crafted description |
| $\mathcal{T}_{\text{vis}}$ | Visual description |
| $\mathcal{T}_{\text{syn}}$ | Synonym description |
| $\mathcal{T}_{\text{PL}}$ | Prompt learning description |
| $\mathcal{D}_{\text{tr}}$ | Train dataset |
| $\mathcal{D}_{\text{te}}$ | Test dataset |
| $\hat{\mathcal{D}}_{\text{cl}}$ | Distinguished clean dataset |
| $\hat{\mathcal{D}}_{\text{no}}$ | Distinguished noisy dataset |
| $\tilde{\mathcal{D}}$ | Re-labeled dataset |
| $g$ | GMM threshold |
| $k$ | GCE parameter |
| $T$ | Learning time |
| $\alpha$ | Relabeling threshold |
| $p_1, p_2$ | GMM Gaussian distributions |
| $G$ | Mean gap between GMM-estimated two distributions |

Table 5: Notations used in the main manuscript.

## B    GENERATING DESCRIPTION FOR ZERO-SHOT CLASSIFICATION

In this section, we describe the way of generating visual description and synonyms in detail.

**Visual description.** In Menon & Vondrick (2023), the authors propose using external knowledge, especially the GPT model, for zero-shot classification of VLMs. They extract the visual characteristics of each class object. For example, in the case of Hen, the visual characteristics of Hen can be summarized as two legs, red, brown, or white feathers, a small body, a small head, two wings, a tail, a beak, and a chicken. When we use those characteristics rather than using human-crafted prompt, such as *A photo of hen.*, it is proven that it improves the zero-shot performance. The main philosophy of this work is to generate various augmented sentences, which are the inputs for VLMs classification. They obtain the description set by following the prompts of the GPT model.

Q: What are useful features for distinguishing a {CLS} in a photo?
A: There are several useful visual features to tell there is a {CLS} in a photo:

Furthermore, the authors of Menon & Vondrick (2023) gave some examples to get the better visual characteristics as follows:

Q: What are useful visual features for distinguishing a lemur in a photo?
A: There are several useful visual features to tell there is a lemur in a photo:

- four-limbed primate
- black, grey, white, brown, or red-brown
- wet and hairless nose with curved nostrils
- long tail
- large eyes
- furry bodies
- clawed hands and feet

Q: What are useful visual features for distinguishing a television in a photo?
A: There are several useful visual features to tell there is a television in a photo:
- electronic device
- black or grey
- a large, rectangular screen
- a stand or mount to support the screen
- one or more speakers
- a power cord
- input ports for connecting to other devices
- a remote control

For getting synonym descriptions, we follow the pipeline of that method. Here is the prompt that we give to GPT model to get the synonyms.

Q: What is the similar words of School bus?
A: There are several synonyms of School bus:
- School transport
- Yellow bus
- School coach
- Student bus
- Educational bus
- Pupil transport
- Children's bus
- School vehicle
- Trolly bus

Q: What is the similar words of Television?
A: There are several synonyms of Television:
- TV
- Telly
- Tube
- Boob tube
- Small screen
- Idiot box
- Cathode-ray tube
- Vid
- Telly
- Receiver

Q: What are the synonyms of {CLS}?
A: There are several synonyms of {CLS}:
-

Here is the example what we obtained using the above synonym extraction prompt.

## C  DATASET AND IMPLEMENTATION

We summarize the datasets what we used in this paper in Table 7. For sample selection for each class, we follow the implementation of Zhou et al. (2022b) and Wu et al. (2023).

| Dataset | Class name | Synonym |
|---|---|---|
| Caltech101 | Motorbike | Bikes, Two-wheelers, Motorized bicycles, Scooters, Mopeds, Motorized cycles, Motorized bikes, Motorized two-wheelers, Motor-driven cycles |
| | Leopard | Jaguars, Pumas, Cougars, Cheetahs, Ocelots, Snow leopards, Clouded leopards, Amur leopards, African leopards |
| Flowers | Pink primrose | Showy evening primrose, Pink evening primrose, Mexican evening primrose, Pink ladies, Buttercups, Sundrops, Pink buttercups, Pink sundrops |
| | Sweet pea | Fragrant pea, Everlasting pea, English pea, Garden pea, Annual pea, Butterfly pea, Winter pea, Spring pea, Summer pea |
| DTD | Cracked | Damaged, Shattered, Fractured, Split, Chipped, Crumbled, Smashed, Flawed, Fault |
| | Grid | Framework, Lattice, Grating, Mesh, Pattern, Structure, Array, System |
| Pets | Havanese | Havanese Silk Dog, Bichon Havanese, Havana Silk Dog, Havanese Bichon, Havanese Cuban Bichon, Havanese Toy Dog, Havanese Bichon Tenerife, Havanese Bichon Havanais, Havanese Bichon Havanueas |
| | Staffordshire bull terrier | Stafford, SBT, Staffie, Nanny dog, Bull and terrier, English staffy, Staffy bull, Staffy dog, Staffy terrier |
| EuroSAT | Highway or road | Expressway, Thoroughfare, Motorway, Route, Street, Lane, Avenue, Boulevard, Byway |
| | Forest | Woods, Jungle, Thicket, Grove, Copse, Timberland, Rainforest, Wilderness, Bushland |
| Aircraft | 737-400 | Boeing 737-400, B734, 737-400ER, 737-400F, 737-400QC, 737-400M, 737-400C, 737-400SF, 737-400 Comb |
| | A310 | Airbus A310, A310-200, A310-300, A310-300F, A310-300MRTT, A310-300C, A310-300QC, A310-300F4, A310-300C4 |
| Cars | Acura TSX Sedan 2012 | Acura TSX 2012, 2012 TSX Sedan, TSX Sedan 2012, Acura TSX Sedan, 2012 Acura TSX Sedan, 2012 Acura TSX Saloon, Acura TSX Saloon 2012, 2012 Acura TSX 4-door, Acura TSX 4-door 2012 |
| | Acura Integra Type R 2001 | Acura Integra Type R 2001 model, 2001 Acura Integra R-Type, Acura Integra R-Type 2001, 2001 Acura Integra R, Acura Integra R 2001 model, 2001 Acura Integra Type R edition, Acura Integra Type R 2001 version, 2001 Acura Integra Type R trim |
| SUN397 | Airport terminal | Air terminal, Terminal building, Airport gate, Departure lounge, Arrival hall, Boarding area, Passenger terminal, Airport hub, Flight terminal |
| | Outdoor athletic field | Playing field, Sports ground, Athletic track, Stadium, Arena, Pitch, Court, Turf, Grounds |
| Food101 | Breakfast burrito | Breakfast wrap, Breakfast taco, Breakfast quesadilla, Breakfast chimichanga, Breakfast roll-up, Breakfast omelette wrap, Breakfast fajita, Breakfast crepe, Breakfast tortilla roll |
| | Carrot cake | Carrot bread, Carrot spice cake, Carrot muffins, Carrot cupcakes, Carrot dessert, Carrot pudding, Carrot torte, Carrot sweet bread, Carrot ginger cake |
| ImageNet | Vulture | Raptorm Carrion birdm Scavengerm Buzzardm Condorm Harpym Kitem Falconm Eagle |
| | Agama | Gecko, Chameleon, Iguana, Dragon, Monitor, Reptile, Salamander, Skink, Anole |
| UCF101 | Biking | Riding, Pedaling, Wheeling, Pedalling, Bicycling, Touring, Spinning, Riding a bike, Cycling tour |
| | Billiards | Snooker, Cue sports, Carom, Pocket billiards, Cue games, Table games, Cue sports, Pocket pool, Carom billiards |

Table 6: Synonym examples obtained from GPT model. For each dataset we desrbie two classes.

| Dataset | Class number | Class example | Description |
|---|---|---|---|
| Caltech101 | 101 | [Airplane, Faces, Motorbikes] | The Caltech 101 dataset is a collection of over 9,000 images distributed across 101 diverse object categories, for benchmarking the object recognition and classification. |
| Flowers | 102 | [Pink primrose, Hard-leaved pocket orchid, Canterbury bells] | The Flowers dataset is a collection of images featuring various flower species, commonly used in the context of image classification and fine-grained flower recognition tasks. |
| DTD | 47 | [Banded, Blotchy, Braided] | The DTD dataset, or Describable Textures Dataset, is a collection of textured images designed for texture analysis in computer vision. It provides a diverse set of textures. |
| Pets | 37 | [Abyssinian, Bengal, Birman] | The Oxford Pets Dataset, also known as the Oxford-IIIT Pet Dataset, consists of images of 37 different fine-grained pet categories, predominantly cats and dogs. |
| EuroSAT | 10 | [Annual crop land, Forest, Herbaceous vegetation land] | The EuroSAT dataset is a collection of satellite images encompassing 10 land use and land cover categories using satellite imagery. |
| Aircraft | 100 | [707-320, 727-200, 737-200] | The Aircraft dataset is a specialized image collection focused on aircraft recognition and fine-grained classification, featuring over 100 aircraft models. |
| Cars | 185 | [AM General Hummer SUV 2000, Acura RL Sedan 2012, Acura TL Sedan 2012] | The Cars dataset is a comprehensive image collection used for fine-grained car recognition, containing over 16,000 images categorized into numerous car models. |
| SUN397 | 397 | [Abbey, Airplane cabin, Airport terminal] | The SUN397 dataset is a large-scale image dataset comprising over 130,000 images across 397 distinct scene categories, valuable for scene recognition and diverse collection of indoor and outdoor scenes. |
| Food101 | 101 | [Apple pie, Baby back rimbs, Baklava] | The Food101 dataset is a collection of over 100,000 images spanning 101 food categories, commonly used for food image classification and recognition. |
| ImageNet | 1000 | [Banded Gecko, Green iguana, Carloina anole] | The ImageNet dataset is one of the most widely recognized and extensive image datasets, containing millions of labeled images across thousands of object categories. |
| UCF101 | 101 | [Apply Eye Makeup, Apply Lipstick, Archery] | The UCF101 dataset is a popular vision dataset with over 13,000 labeled action video clips spanning 101 human action categories, commonly used for action recognition research. |

Table 7: Dataset Description

# D   HOW TO GENERATE NOISY DATASET

Different from conventional noisy label papers, this paper tries to examine the impact of noisy labels on VLMs trained with few-shot images. Therefore, we summarize how we construct noisy labels in both symmetric and asymmetric cases.

**Symmetric.** The most basic case involves symmetric noisy labels. We generate symmetric noisy labels using the following steps in the few-shot case: (1) First, select 16-shot images for each class. These samples form the training dataset $\mathcal{D}_{\text{tr}}$. (2) Then, select noisy label candidates with a given per-class noisy ratio among the 16 samples and flip their label to one of the remaining classes. For example, in the case of the $c^{\text{th}}$ class, it is flipped to the others uniformly at random, *i.e.,* $\hat{y} \in \{1, \ldots, C\} \setminus \{c\}$.

**Asymmetric.** Different from the prior work Wu et al. (2023), we first examine the asymmetric case. Following prior works such as Ko et al. (2023) in robust training method trained from scratch, we first select half of the classes, $\hat{\mathcal{C}} \subset \{1, \ldots, C\}$, where $|\hat{\mathcal{C}}| = \lfloor \frac{1}{2} \times C \rfloor$. We then generate a matching between $c \to c' : c \in \hat{\mathcal{C}} \to c' \in \{1, \ldots, C\} \setminus \hat{\mathcal{C}}$. We select samples in each class $c$ with the given ratio and change their labels to the mapped class $c'$. This means that if the ratio is $50\%$, then the number of samples in $c$ is 8, while the number of samples in $c'$ is 24, comprising 16 clean and 8 noisy samples. This case indicates that the noisy ratio in $c'$ is severe.

# E    THE PORTION OF THE SELECTED PROMPT

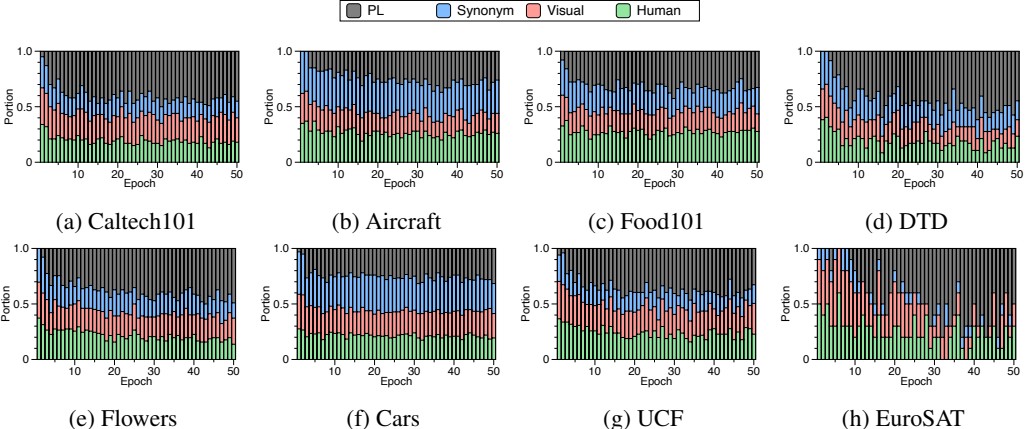

Figure 7: Selected portion of each prompt when we run PoND.

We describe the portion of each prompt being selected as epoch goes on. Note that one zero-shot prompt cannot be selected in the next epoch due to the randomness of GMM. As described in Figure 7, at the beginning of each training, PL does not have enough portion which means zero-shot knowledge is used for selecting noisy samples. Afterwards, the portion of PL smoothly increases, while the others decreses. It means that the trained knowledge occupies the other's role as training goes on.

# F    OTHER TYPES OF PROMPT LEARNING.

| Method | Caltech | EuroSAT | Flowers | Pets | Cars | DTD | Food | Average |
|---|---|---|---|---|---|---|---|---|
| MaPLe (Clean) | 97.53 | 84.71 | 87.64 | 95.37 | 65.66 | 68.68 | 90.69 | 84.33 |
| MaPLe | 90.34 | 47.45 | 42.13 | 82.66 | 51.96 | 56.84 | 83.75 | 65.02 |
| MaPLe + PTNL | 97.37 | 56.85 | 80.24 | 94.83 | 63.62 | 56.81 | 89.54 | 77.04 |
| MaPLe + PoND | 97.48 | 66.47 | 83.55 | 95.16 | 64.17 | 62.36 | 90.20 | 79.91 |
| PromptSRC (Clean) | 98.36 | 94.13 | 97.88 | 95.39 | 79.50 | 80.92 | 90.57 | 90.96 |
| PromptSRC | 98.37 | 74.34 | 84.21 | 86.34 | 57.35 | 57.36 | 79.88 | 76.84 |
| PromptSRC + PTNL | 97.96 | 70.35 | 80.88 | 95.23 | 68.80 | 75.19 | 90.41 | 82.69 |
| PromptSRC + PoND | 98.15 | 71.34 | 81.53 | 95.24 | 73.40 | 77.35 | 90.42 | 83.92 |

Table 8: Other Prompt Learning with noisy labels. We test on $50\%$ symmetric noisy labels on Seven datasets. We report ViT model's performance, since they support ViT model only.

We check the performance of the most recent PL method on VLMs, i.e., MaPLe Khattak et al. (2023a) and PromptSRC Khattak et al. (2023b). We directly modify the official implementation of PromptSRC, which supports MaPLe as well. As described in Table 8, when noisy labels are injected into the training dataset, especially in the seen class, both previous algorithms suffer from performance degradation. When we utilize the proposed algorithm, it works well with other types of PL methods compared to the PTNL.

# G    PERFORMANCE ON THE CLEAN DATASETS.

| Dataset | Vanilla | PTNL | Ours |
|---|---|---|---|
| DTD | 66.19 | 66.01 | 77.20 |
| Caltech101 | 92.47 | 92.56 | 92.61 |
| EuroSAT | 77.68 | 77.71 | 77.50 |
| Flower | 90.40 | 90.26 | 90.23 |
| Cars | 68.99 | 68.61 | 69.19 |
| Aircraft | 28.95 | 28.90 | 28.92 |

Table 9: Performance on the datasets without noisy labels.

We have measured the performance of Vanilla, PTNL, and PoND in the absence of noisy labels. As shown in Table 9, the three algorithms exhibit comparable performance. This indicates that while any

algorithm may suffice in the absence of noisy labels, the proposed PoND algorithm should be used when noisy labels are present.

## H    OTHER SAMPLE SELECTION STRATEGY

| Dataset | GMM | Random | Lowest Loss |
|---|---|---|---|
| DTD | 63.12 | 59.69 | 60.71 |
| Caltech101 | 95.14 | 92.43 | 93.52 |
| EuroSAT | 74.58 | 73.14 | 73.52 |
| Flower | 88.47 | 86.70 | 87.72 |
| Cars | 67.51 | 65.21 | 66.92 |
| Aircraft | 28.73 | 26.52 | 27.52 |

Table 10: Performance on other selection strategies.

As shown in Table 10, through additional experiments comparing the two proposed prompt selection methods, we confirmed that the current proposed method, which selects prompts using GMM, is superior. We compared two additional methods: random prompt selection (Random) and lowest loss prompt selection (Lowest Loss). As shown in the experimental results below, performance improves as the strategy is updated. Therefore, we argue that the proposed GMM-based prompt selection method is more effective.

## I    OTHER SAMPLE SELECTION STRATEGY

| Algorithm | GMM |
|---|---|
| Vanilla (CoOp) | 4m 7.725s |
| PTNL | 4m 14.607s |
| PoND (Ours) | 5m 41.95s |

Table 11: Training cost analysis.

As shown in Table 11, we conducted an analysis of the computational cost. The required experiment time for the DTD dataset is provided below. As shown in the table, although PoND incurs a higher cost compared to PTNL or Vanilla, it only requires a relatively short time (5 minutes), indicating that the cost is not significant. This demonstrates that PoND leverages the advantages of prompt learning to provide a robust learning method against noisy labels.

