# OpenReview forum: "Robust Prompt Learning For Vision-Language Models With Noisy Labels"
_ICLR.cc/2025/Conference — Submitted to ICLR 2025_

### Official Review · Reviewer_4JGY · 2024-10-27

**Soundness:** 3
**Presentation:** 3
**Contribution:** 2
**Rating:** 5
**Confidence:** 4

**Summary:**

Vision-Language Models (VLM) have shown remarkable performance in visual scene understanding tasks especially image classification in downstream datasets. However, for a good performance, having expert annotated cleanly labeled datasets are necessary. This paper assumes downstream datasets have noisy labels. The authors, first, examine the behavior of pre-trained VLMs under three different types of textual prompts – human crafted, using synonyms and using a Large Language Model (LLM) generated description of the classes. They found that all these prompts behave distinguishably between the clean and the noisy samples. So, the authors find which of these three types of prompts gives the maximum separation between two clusters formed by the cross-entropy losses of all the examples for each class. These class specific expert prompts provide the clean and the noisy samples. For the noisy samples, a relabeling is done using the remarkable zero-shot classification capability of the VLMs. With the clean and relabeled samples, the VLM is adapted to the downstream tasks using prompt learning.

**Strengths:**

1.	The analysis done by the authors using different types of prompts on their capability of distinguishing between clean and noisy labeled samples is good. It provided the insight that a good prompt can indicate by virtue of the loss values on the samples, which samples may possibly be noisy. This first label filtering helps get a strong set of candidates for relabeling.
2.	The baseline analysis of directly assigning inference label to all samples, threshold driven assignment of inference label from different types of prompts to samples having different noisy labels are also good.
3.	The ablation experiments including number of given images per class, position of the prompts, number of prompts and hyperparameter sensitivity analysis adds to the experimental strength of the paper.

**Weaknesses:**

1.	It seems, the authors went ahead with only synthetic label noise injected into 11 datasets. However, previous works on Learning with Noisy Labels [Karim et al. (2022), Li et al. (2020b)], have additionally showed their performance on real-world large-scale noisy datasets like Clothing1M, WebVision etc. Without comparison with existing noise robust approaches in these naturally noisy datasets, the superiority of the proposed approach can not be established.
2.	One of the crucial steps of the proposed approach is to detect the samples with noisy labels effectively before relabeling. However, there isn’t any evaluation of performance on this step. It would have been good the know the precision-recall performance of the prompt based ‘Set_Distinguishing’ step where a good precision would mean greater proportion of actual clean samples in $\mathcal{D}_{cl}$ and a good recall would mean more clean samples are identified from the noisy dataset.
3.	Unlike existing prompt tuning approaches like [Khattak et al. (2023a), Wu et al. (2023)] the proposed approach did not use the VIT-B/16 or VIT-B/32 backbone for experimentations. Having such experiments would have enabled a better comparison with other approaches, not necessarily in learning with noisy labels only.
4.	Minor typo: Line 363-364: ‘are use’ -> ‘are used’. Line 393: ‘slighter’ -> ‘less’.

**Questions:**

1.	How is the comparative performance with existing approaches for Learning with Noisy Labels in non-synthetic noisy data? Refer to weaknesses (Point 1) for more details.
2.	How is the performance of the ‘Set_Distinguishing’ step in getting noisy and clean labels? Refer to weaknesses (Point 2) for more details.
3.	Why are results in more common backbones VIT-B/16 or VIT-B/32 not given?
4.	Line196-197: How is ANT_C chosen?
5.	The figure and table captions could have been more informative. For example, what is the evaluation metric in Fig. (2)? What do $S$ and $A$ mean in the column headers of Table 1?

---

> ### Comment · Reviewer_4JGY · 2024-12-03
> **Lack of author response**
>
> Without the response from the reviewers, I shall keep my original rating for the submission.

---

### Official Review · Reviewer_ZtEK · 2024-10-30

**Soundness:** 2
**Presentation:** 3
**Contribution:** 2
**Rating:** 5
**Confidence:** 4

**Summary:**

This paper introduces a robust training method called PoND, which utilizes a complementary approach across various types of prompts. The authors also explore the behavior of pre-trained VLMs under different prompts, including those crafted by humans and LLMs with visual characteristics. Lastly, the proposed PoND method outperforms existing prompt learning methods with noisy labels (e.g., CoOp and PTNL).

**Strengths:**

1. The proposed method PoND introduces a novel robust training process into prompt learning with noisy labels. Specifically, PoND consists of three steps for each iteration: distinguishing the _clean_ and _noisy_ samples in a training iteration (epoch), re-labeling noisy labels, and using GCE loss to traing models. Moreover, the good experiment results verify the proposed method. Besides, the experiment analysis and ablation study are sufficient.

2. I like the tables and figures of this paper. They are clear and.

**Weaknesses:**

The performance of PoND is impressive compared to the current state-of-the-art approach, PTNL. However, its primary weakness is the computation cost because it need additional steps to clean or denoise noisy labels. As shown in Table 11, the training times for CoOp, PTNL, and PoND on the DTD dataset are 4m 7.725s, 4m 14.607s, and 5m 41.95s, respectively. This indicates that PoND's training time is over 38% longer than Vanilla, which could hinder its broader application. And it would be beneficial to provide more details on computational costs for large datasets like ImageNet.

Furthermore, two of the three proposed steps in PoND are existing methods, namely GMM estimation and GCE loss. Thus, the contribution may not be substantial enough for the whole community.

**Questions:**

Can the proposed method be extended to other applications of prompt learning, such as segmentation or even NLP tasks?

---

### Official Review · Reviewer_K8Kb · 2024-11-02

**Soundness:** 2
**Presentation:** 2
**Contribution:** 2
**Rating:** 5
**Confidence:** 4

**Summary:**

The paper introduces PoND, a prompt learning method for vision-language models under noisy labels. The paper first analyses the effectiveness of various prompts (human-crafted, synonym-based, description-based) for zero-shot classification, revealing that each prompt has specific strengths across classes. Building on this, PoND classifies samples as clean or noisy using Gaussian mixture models and assigns pseudo-labels to noisy samples with high confidence. Experimental results across 11 datasets show PoND's good performance in maintaining robustness under noise compared to existing methods.

**Strengths:**

1. The discovery that different prompts—Human, Synonym, and Description—each excel in specific classes under noisy labels is interesting and adds depth to prompt engineering in vision-language models.

2. The idea of PoND is interesting, which combines prompt expertise with GMM and high-confidence pseudo-labeling, demonstrating a robust approach to noise handling.

3. Extensive experiments across 11 datasets confirm PoND’s superior performance, especially in high-noise scenarios.

**Weaknesses:**

1. The paper spends significant space demonstrating that different prompts—human-crafted, synonym-based, and description-based—offer distinct advantages across datasets and categories, which is intuitive and already discussed in prior works; however, the implementation and advantages of each module in the proposed method are less clearly detailed. Also the choice of the three selected prompts seems somewhat ad hoc.

2. In the first module of the method (set distinguishing), the authors do not provide comparative results between different expert models, only presenting results without this module.

3. In the re-labeling step, the authors rely on CoOp to generate pseudo-labels but do not attempt comparisons with newer prompt learning methods, e.g. Shuvendu Roy and Ali Etemad. Consistency-guided prompt learning for vision-language models. ICLR 2024.

4. The paper lacks comparative experiments on clean datasets, such as using only the expert model in Module 1 for direct prediction results.

5. Paper writing is sloppy, e.g. it uses a non-standard citation style, table conventions are specified in Appendix D making each individual table a hard to parse puzzle, the tiny font also does not help.

6. The Related Work section is unfocused and does not highlight the novelty of the proposed method well, especially in light of prior prompt learning methods and methods for learning with noisy labels. Also note LNL is used but never properly defined. A division into two topics (prompt learning and learning with noisy labels) would make the section much easier to consume.

**Questions:**

My main questions have been outlined in the weaknesses, with primary concerns centered around missing comparative experiments. If the authors can provide these additional comparisons, I will reconsider my position for this paper. Additionally, here are some further questions:
1. How does the observed effect of using different prompt types perform on traditional few-shot prompt learning datasets?
2. When using synonyms, how many synonym words were included in the experiments? The same question applies to descriptions.
3. In the figures, "Visual" should be labeled as "Description" to avoid potential misunderstanding.

Note that while I am rating the paper currently as 5, I would have chosen 4 if that was an option.

---

### Official Review · Reviewer_keRF · 2024-11-03

**Soundness:** 3
**Presentation:** 2
**Contribution:** 2
**Rating:** 5
**Confidence:** 4

**Summary:**

The paper tackles the issue of noisy labels in datasets when adapting pretrained Vision-Language Models (VLMs) to target domains with noisy training data. It presents a method using prompt techniques to differentiate between clean and noisy data, relabeling the latter based on confidence predictions. The relabeled data is combined with clean data to train domain-specific learnable prompts, aiming to improve classification accuracy.

**Strengths:**

The paper addresses the practical challenge of noisy training data in image classification tasks.

The study explores the effects of various prompts and introduces synonyms as a novel data augmentation technique.

Detailed ablation studies are conducted to verify the effectiveness of the proposed method.

**Weaknesses:**

1. The overall contribution is limited. The effectiveness of using synonyms is intuitive, and correcting mislabeled samples is not novel [1]. The use of four types of prompts to identify incorrect labels, claimed to be better than zero-shot identification, resembles prompt ensemble. A comparison study with simple prompt ensembles should be included.

2. The experiments should be significantly extended. It's surprising to see only two methods compared in the experiment section (CoOp and PTNL). More methods should be included, such as CLIP pseudolabeling methods [2], traditional denoising methods applied to CLIP [1, 3], and advanced CLIP transfer learning methods [4, 5].

3. The additional cost (e.g., inference time, monetary cost) of querying GPT is not discussed, which cannot be ignored.

4. The "syn" prompt (Line 200) is confusing. Since CLIP is known for its bag-of-words feature [6], it may not distinguish "This is a photo of {CLS}. It is not a {ANT}." from "This is a photo of {ANT}. It is not a {CLS}.", raising doubts about the method's effectiveness.

5. Citations should be in parentheses (use \citep{}) when the authors or publication are not included in the sentence.

6. The paper is somewhat hard to follow. For example, the introduction lacks sufficient information for readers to understand the complex teaser figure.

[1] Learning from Massive Noisy Labeled Data for Image Classification. CVPR 2015.

[2] Enhancing CLIP with CLIP: Exploring Pseudolabeling for Limited-Label Prompt Tuning. NeurIPS 2023.

[3] CleanNet: Transfer Learning for Scalable Image Classifier Training With Label Noise. CVPR 2018.

[4] Self-regulating Prompts: Foundational Model Adaptation without Forgetting. ICCV 2023.

[5] Not All Features Matter: Enhancing Few-shot CLIP with Adaptive Prior Refinement. ICCV 2023.

[6] When and Why Vision-Language Models Behave like Bags-Of-Words, and What to Do About It? ICLR 2023.

**Questions:**

1. In L303, pseudo-labels are generated using predictions from initialized prompts. Does it make sense for the model to learn from the labels it predicts?

2. How was the loss in Figure 3 calculated?

---

### Official Review · Reviewer_8daF · 2024-11-12

**Soundness:** 2
**Presentation:** 3
**Contribution:** 2
**Rating:** 5
**Confidence:** 4

**Summary:**

This paper enhances classification fine-tuning performance by leveraging the zero-shot classification capability under a noisy labeled training dataset. Specifically, authors introduce a new training method called PoND, which employs a complementary approach across different types of prompts, leveraging the expertise of each class. The proposed method outperforms prior approaches across 11 real-world datasets.

**Strengths:**

- The paper is well written and easy to follow.
- The results provide the valuable hints that what's the better way to deploy the pretrained vision-language model.
- The paper inspire people to think about more effective prompt design.

**Weaknesses:**

- Although the paper states “Vision-Language Models” in its title, the experiments are only performed on CLIP models. It would be great to see similar findings for other vision-language models like DeCLIP, FILIP, CLOOB, CyCLIP, etc. Given that the paper is more of an analysis paper instead of a methodology paper, I would expect authors to verify their claims on other VLMs. Does the size of models effect the conclusions presented in this paper? A more thorough comparison and analysis should be included in the paper.
- How is this paper different from Rethinking the Value of Prompt Learning for Vision-Language Models? Isn't the conclusion opposite to this paper? It would be good discuss this paper in the lens of prompt learning vs classifier finetuning for VLMs.
- How does the proposed method work with respect to distribution shifts across domains, e.g., applying POND in ImageNet while testing on ImageNet-V2 or ImageNet-Sketch?

**Questions:**

I think the identified problem is important but I’d like to rate the current submission below the acceptance threshold due to limited technical contributions and lack of convincing experiments. The paper needs significant changes including new experiments and possibly methodological improvements in justifying the motivation/real use behind the proposed method.

---

### Meta-Review · Area_Chair_h1Vy · 2024-12-17

**Metareview:**

The five reviewers unanimously gave borderline reject rating. The reviewers noted that the method has limited novelty and has similar idea to prompt ensembling, which however was not compared in the experiments. They also mentioned the lack of comparison with other relevant methods and the omission of details on computational costs and effectiveness under distribution shifts. Additionally, the paper's clarity and structure need improvement. The authors did not submit their rebuttal.

**Additional Comments On Reviewer Discussion:**

No rebuttal was submitted.

---

### Decision · Program_Chairs · 2025-01-22

Reject